# Modeling non-pharmaceutical interventions in the COVID-19 pandemic with survey-based simulations

**Marius Kaffai** [ID]*, **Raphael H. Heiberger**

Institute for Social Sciences, University of Stuttgart, Stuttgart, Germany

* marius.kaffai@sowi.uni-stuttgart.de

**Data Availability Statement:** The simulation output data and the code to reproduce the plots and result tables are stored in this repository: DOI 10.17605/OSF.IO/B9YPG. Unfortunately, we are not allowed to share the Socio-Economic Panel

## Abstract

Governments around the globe use non-pharmaceutical interventions (NPIs) to curb the spread of coronavirus disease 2019 (COVID-19) cases. Making decisions under uncertainty, they all face the same temporal paradox: estimating the impact of NPIs before they have been implemented. Due to the limited variance of empirical cases, researchers could so far not disentangle effects of individual NPIs or their impact on different demographic groups. In this paper, we utilize large-scale agent-based simulations in combination with Susceptible-Exposed-Infectious-Recovered (SEIR) models to investigate the spread of COVID-19 for some of the most affected federal states in Germany. In contrast to other studies, we sample agents from a representative survey. Including more realistic demographic attributes that influence agents' behavior yields accurate predictions of COVID-19 transmissions and allows us to investigate counterfactual what-if scenarios. Results show that quarantining infected people and exploiting industry-specific home office capacities are the most effective NPIs. Disentangling education-related NPIs reveals that each considered institution (kindergarten, school, university) has rather small effects on its own, yet, that combined openings would result in large increases in COVID-19 cases. Representative survey-characteristics of agents also allow us to estimate NPIs' effects on different age groups. For instance, re-opening schools would cause comparatively few infections among the risk-group of people older than 60 years.

## Introduction

In 2020, governments tried to curb the spread of COVID-19 without having an appropriate medical response available. Most governments implemented non-pharmaceutical interventions (NPIs) which affect societal life in an unseen manner. Although there is initial evidence that NPIs, such as the closure of educational facilities [1–3], workplaces [3, 4] and certain businesses [1, 2], the cancellation of mass gatherings [1–3], case detection and contact tracing [4, 5], shielding of vulnerable people [6], mobility restrictions [7], wearing masks [8] or specific network intervention strategies [9] are effective in reducing the amount of new cases, it is also clear that NPIs come with severe side effects, e.g., on the economy [9–11], labour markets [12],

(SOEP) data or parts of it due to a very restrictive data access policy. Therefore, we cannot share data on the agent population based on the SOEP. The SOEP is a representative panel study for Germany conducted annually by DIW Berlin. Researchers affiliated with an institution can request access to SOEP at this website: https://www.diw.de/en/diw_01.c.601584.en/data_access.html. All of the code used to run the simulation and the publicly available datasets used to calibrate the simulation model are available in this repository: https://github.com/mariuzka/covid19_sim.

**Funding:** The authors received no specific funding for this work.

**Competing interests:** The authors have declared that no competing interests exist.

people's physical [13] and mental health [14], or concerns on increasing domestic violence [15]. Thus, decision-makers act under immense pressure to balance public health risks while sustaining economic activities. In so doing, policy-makers are facing a temporal paradox when it comes to the implementation of NPIs: they have to decide whether a particular NPI (or a set thereof) is curbing the spread of COVID-19 without having reliable knowledge about the NPIs' actual effectiveness. The best way to evaluate the effectiveness of NPIs would be to measure the empirical correlation between the implementation of a NPI and the change of the infection rate. However, an empirical estimation is difficult in many areas due to the unavailability of data. Only few studies exist with a rather narrow set of scenarios [1–3] or coarse resolutions [16]. Empirical studies on the effect of NPIs are by-design limited in scope because they can only measure the impact of what governments implement, i.e., estimations rest on scenarios which have actually happened. Based on inter-temporal cross-country data, Brauner et al. [1], for instance, cannot disentangle the effect of school closure and the effect of university closure, because in most countries these two institutions were closed almost at the same time. To circumvent one's reliance on detailed empirical data in order to evaluate a larger variety of scenarios, simulation methods can be used to create and investigate the effect of NPIs which did not occur in reality [17, 18].

In addition, there is great need for more detailed knowledge of how NPIs work in the specific context of a country's or region's social structure [17, 19]. Preliminary evidence exists that the spread of COVID-19 cases is starkly influenced by structural circumstances, such as age distributions [20, 21] or network densities of geographic areas [22]. To reflect social structures and relevant characteristics of people's behavior in simulations, agent-based modeling (ABM) have proven to be good complements to epidemiological differential equation models [9, 17, 23, 24]. Once a model with predictive capability is established [25], it is possible to create scenarios that differ from empirical scenarios and to apply the simulation results to reality [17, 18].

Many researchers already exploit the benefits of data-driven agent-based models to investigate NPIs in the context of a countries' specific circumstances [4, 6, 26–29]). ABM exist for the U.S. [26, 30], Canada [4], Colombia [31], Australia [27], France [6], Singapore [28], Iran [29] and the UK [26]. Our model not only adds (parts of) Germany to this list but uses an innovative method to generate an artificial population of agents which resembles the real population of a federal state and, hence, important parts of its social structure. Thus, instead of corresponding to a few, rather coarse macro-characteristics of the target population, our study represents households of survey participants. The micro-level of the simulation—agents with specific attributes living together with other agents with specific attributes—therefore reflects important properties of the social structure of a society; much more than synthesizing the artificial population based on aggregated macro-data on the target populations's household structure.

## Materials and methods

The overall purpose of our simulation study is to investigate the effects of various non-pharmaceutical interventions (NPI) in curbing the spread of COVID-19. We predict case numbers of counter-factual scenarios in which certain NPIs are (not) implemented and compare the predicted case numbers to empirical case numbers. We evaluate our agent-based model by its ability to reproduce time-series data of observed case numbers in a baseline scenario that mimics the factual implementation of NPIs happened in spring 2020 in Germany. Hence, dropping certain measures or combinations thereof (i.e., deviations from the empirical baseline scenario) allows us to estimate each NPI's efficacy. We model the infection on the level of federal

states. We chose the four states in Germany (Baden-Wuerttemberg, Bavaria, Hamburg, Saarland) with the highest infection rate relative to its population size during the first wave of Covid-19 in Germany (March 2020). To describe the model, we roughly follow the ODD (Overview, Design concepts, Details) protocol for describing individual- and agent-based models [32, 33]. All code is made available to reproduce results or to add further scenarios and, respectively, NPIs.

## Entities and scales

The four main entities driving the simulation's dynamics are agents, locations, a virus and a global environment. The agents represent humans living their daily life in the pandemic. During the day, they execute various activities and spend their time either at home, kindergarten, school, university, supermarket or an occupation-specific work place. Each action depends on the day of week, the time of day and individual characteristics of each agent. The values of the agents' characteristics, such as age, gender, occupation, daily working hours or the time usually spent in the supermarket, are taken from survey data, i.e., each agent represents a survey respondent in regard to his or her attributes.

Locations represent the different sites where agents can reside during the day and between which agents move. Space is modeled by locations, however, no geographical distance between different locations are represented. The types of locations implemented in the model are agents' homes, workplaces, school classes, kindergartens, universities and supermarkets, in order to simulate the most important NPIs that had come into effect in each federal state in Germany in Spring 2020. Whenever possible, the number of each type of locations is determined by the composition of the agent population and empirical macro data (cf. subsection Initialization). Locations can only be accessed by assigned agents, e.g., only members of a specific household can enter their home or only employees of a specific company can enter the company's office.

The virus represents COVID-19 infecting the agents. If an agent gets infected by the virus, a multistage disease process is triggered. The virus can be transmitted from an infectious agent to a susceptible agent, if these two are at the same location (cf. subsection Disease process and virus transmission).

The global environment represents laws and decrees influencing the daily schedule of the agents (cf. subsection Non-pharmaceutical interventions). This means especially the implementation of NPIs, but also the closing of schools due to state-wide holidays. For the base model we create a schedule of NPI implementations and relaxations that mimics the history of events during the first wave of COVID-19 in Germany. In the simulation experiments, we then design contra-factual scenarios using schedules that differ from the empirical baseline scenario (cf. subsection Scenarios and model execution).

Time is represented via discrete steps. 18 time steps represent one day, where each time step usually represents one hour. Only in the night time of each day 6 hours are skipped due to run-time optimization. We assume the night time when most people sleep is not relevant for the spread of the virus. Each run simulates 100 days (1800 time steps), starting with the day when the case numbers in the federal state are closest to 50 cases per 100,000 inhabitants. (Start dates and end dates: Hamburg 2020.03.19–2020.06.26; Baden-Wuerttemberg: 2020.03.21–2020.06.28; Bavaria: 2020.03.23 -2020.06.30; Saarland 2020.03.25–2020.07.02).

## Initialization

The first step of model initialization is to create the population of empirical-grounded agents. Each agent-based model includes a population of at least 100,000 agents. The values of the

agents' characteristics, such as age, gender, occupation, daily working hours or the time usually spent in the supermarket, are taken from survey data, i.e., each agent represents a survey respondent in regard to his or her attributes. For that purpose, we use the 2017 German Socio-Economic Panel (SOEP) [34]. The SOEP is a representative panel survey that includes entire households of respondents. To set the agents' daily routines in the simulation, we classify each agent either as kindergartner, school student, university student, working university student, working or non-working agent. All agents in the age from 0 to 5 are classified as kindergarten kids. Each agent in the age from 6 to 19 is classified as school kid. Older agents are either university students, working university students, working (with a certain occupation) or non-working agents. In order to distinguish the occupational fields of the agents we use the current version of the Statistical Classification of Economic Activities in the European Community (NACE Rev. 2) on level 1 (NACE-sections) and level 2 (NACE-divisions) provided in the SOEP. S1 Table lists all variables used to set agents' attributes.

Like every survey, SOEP has to deal with missing values. For 3.53% of the sample we replace missing data on working hours by age- and gender-specific group means. For 3.46% of the sample we replace missing data on the occupation by a random occupation code in each simulation run. For cases with missing information about the time spent daily in supermarkets (26,92%) we impute the overall mean (approximately 1h). Finally, 1.62% of the households are excluded, because the case itself or a member of the corresponding household has missing data.

To create a population of agents that represents a given federal state, we first create a subset of SOEP-respondents who reside in the given state. In a second step, we take a random sample of households (with replacement) from the subset until a population size of at least 100,000 is reached. The households' selection probabilities are not equal, but are weighted by the corresponding cross-sectional representativity weight provided by the SOEP for each household. Due to the random sampling of households of different sizes, the final population size is either exactly or slightly above 100,000.

After the creation of the population, the simulated world is built, which consists of different types of locations. In a first step, for each type of location the necessary frequency is calculated. In a second step, the agents are assigned to selected locations. For each household of agents a home is created in which all members of the household (taken from the SOEP) reside. The number of additional location types is adjusted to the characteristics of the agent population and the respective federal state. S1 Table shows the parameters we use to calculate the corresponding number of locations in the simulation model. After all locations are created, kindergarten kids are randomly assigned to one of the kindergartens, pupils are randomly assigned to one of the school classes, university students are assigned to the university and working agents are randomly assigned to one work place of the corresponding NACE-section. All agents above the age of 13 are assigned to two randomly selected super markets.

In order to start a pandemic when the simulation executes, at the beginning of each single simulation run the infection status of 50 randomly selected agents is set to infectious.

## Main simulation loop

After the initialization, the main simulation loop starts and simulates one (hourly) time step every round until 100 days were simulated. In each time step, the current date and time are updated by incrementing the datetime by one hour. Because most of the night time is skipped, the clock is set forward to 7 a.m. if it is 1 o'clock.

Every agent starts each time step by an update of its infection status, i.e., an agent checks whether it has to set it's infection status from exposed to infectious with symptoms or from

infectious with symptoms to recovered (cf. subsection Disease process and virus transmission). In the next step, if the agent is infectious, it potentially infects (susceptible) agents present at the same location (cf. subsection Disease process and virus transmission). In the next two processes the agent first checks whether it stays at home that day due to feeling sick or due to an ongoing isolation of the whole household (cf. subsection Non-pharmaceutical interventions). If neither is the case, agents start, continue or finish an activity depending on the weekday, the clock time, the age, student status, daily work hours and the NPIs that are currently in effect (cf. subsection Activities).

## Activities

Activities cause agents to leave their home during the day, to visit other locations, to encounter other agents, and as a consequence, to potentially spread the virus through physical proximity. An activity is implemented in the simulation as staying at a location for a certain period of time. Every agent has a daily schedule of activities that determines which locations an agent visits, and for how long. The default activity that every agent exhibits when there is no other activity on its schedule is being at home. From Monday to Friday at 8 o'clock, kindergarten kids go to kindergarten, school kids go to school, university students attend university, and workers go to work. School kids and kindergarten kids return to their homes after 5 hours, university students after 4. While the dwell time at kindergarten, school and university are based on assumptions, the average daily working hours represent those of the modeled survey respondents (we have tested the effect of differing assumptions on the case numbers in the baseline model—cf. S1–S3 Figs). If university students have jobs, they first go to work and then go to university. When agents are at home and there is no other activity, there is a certain probability to go to the supermarket. The average time an agent spends at the supermarket is taken from the SOEP. To account for the possibility of home office in the simulation, working agents change their location to their workplace only with a certain probability and otherwise start working from home. The probabilities to work from home are derived from data on the frequency of home office use per NACE-division in Germany in 2018 [35] (cf. S4 Table).

## Disease process and virus transmission

The biological aspect of the infection (i.e., what happens to an agent after the virus was transmitted) was implemented in strong orientation to the way the process of infection is implemented in Covasim [36]. It can be considered as an extended S-E-I-R (susceptible, exposed, infectious, recovered) infection model [37]. The process of infection by COVID-19 is modeled by an multi-stage infection process including up to six different phases of infection an agent can have: susceptible, exposed, infectious, infectious with symptoms, infectious without symptoms and recovered. Except for 50 randomly selected agents starting with the status infectious, every agent is initially susceptible to an infection. When an agent got infected, the agent goes through the stages of infection. For each agent and each stage of infection we draw individual time spans from a log-normal-distribution following Covasim [36] (cf. S1 Table). In addition, the decision whether an agent develops symptoms or not is determined by an age-dependent probability, which also resembles Covasim [36].

The first stage after the infection occurs is being exposed to the virus. At this stage, the virus has entered the agent's body, but the agent is not yet infectious or symptomatic. The second stage of infection is being infectious. This is the time during the infection when the agent is already infectious but not yet symptomatic because the body has not had enough time to develop symptoms. In the next stage the agent can either develop symptoms or remain completely asymptomatic. Finally, an agent reaches the stage of being recovered from the

infection. We assume permanent immunity after recovery, thus, an agent cannot be infected for a second time.

The agent-to-agent transmission of the virus is one of the most essential elements in the modeling of infectious diseases. In our model, a transmission of the virus can occur whenever an infectious agent and a susceptible agent are at the same time in the same place. At each time step, each infectious agent randomly selects one agent that is currently in the same location (if the infectious agent is currently not alone in its location). If the agent randomly selected by the infectious agent is susceptible to infection, its infection status is set to 'exposed' by a certain probability of infection. This infection probability per time step is an important, yet unknown parameter and is therefore determined by model calibration to empirical infection data of a given state (cf. section Calibration). In the model calibrated for this study, these infection probabilities range from 0.04727 to 0.0645 (cf. Table 3).

## Non-pharmaceutical interventions

Non-pharmaceutical interventions (NPIs) affect the agents' daily schedule by canceling certain activities and causing the agents to stay at home instead. The changes on the agents' daily lives due to NPIs considered in the model are the following: less to no time at school, kindergarten or university, an increased amount of time working from home, quarantining households of infected agents and reduced or no work hours due a (partial) shutdown or economic downturn in certain occupational fields. For most of the NPIs a daily probability distribution determines for each agent individually whether it starts a certain activity and leaves its home or not. In response to active NPIs, the specific probability reflects whether a certain location is closed (or to what degree), or that work hours are reduced for a given NACE-section. For example, a (partially) closed school is implemented as a daily probability that determines for each school kid separately whether he or she goes to school that day. To model that only half of the pupils are allowed to go to school, the daily probability to go to school for each pupil is set to 0.5. If schools are completely open, the probability for school attendance is 1. Complete closure of schools are reflected in a daily probability of school attendance equal to 0. If, for instance, 80 percent of a specific NACE-division has to work from home, the daily probability to visit the work place is set to 0.2 for all agents belonging to that specific NACE-division. Besides the potential closure of work places and educational facilities also the isolation of households is implemented as an NPI. If an agent shows symptoms of an infection over a certain period of time, the agent and all household members are quarantined at home for 14 days. The period of time from the onset of symptoms of one of the household member to the isolation of the whole household is one of the two calibrated parameters and lies between 21 and 26 time steps (cf. Table 3).

## Scenarios and model execution

The model allows us to simulate different scenarios where the duration and degree of implementation of specific NPIs can be varied by entering specific *timetables* during model initialization. Those timetables contain daily information about the state of each NPI. Because the initial goal of the model is to capture the course of the pandemic during its first wave in Germany, we create a baseline model with a timetable which mimics the factual implementation of NPIs as it took place in Germany in spring 2020. In the baseline model we mimic the course of the lockdown in Germany with its onset at around march 16 and the following 100 days of disease spread. Table 1 shows the varying degree of implementation of different NPIs by date in the baseline scenario.

**Table 1. Timetable of NPIs in baseline scenario.**

| NPI | March 16 | April 20 | May 6 | May 18 | June 2 | June 15 |
|---|---|---|---|---|---|---|
| Kindergartens | 0.1 | 0.1 | 0.3 | 0.3 | 0.3 | 0.6 |
| Schools | 0 | 0.1 | 0.3 | 0.3 | 0 | 0.3 |
| NACE-section G | 0.25 | 0.5 | 1 | 1 | 1 | 1 |
| NACE-section I | 0.25 | 0.25 | 0.5 | 1 | 1 | 1 |
| NACE-section P | 0.25 | 0.5 | 1 | 1 | 1 | 1 |
| NACE-section R | 0.25 | 0.5 | 0.75 | 0.75 | 0.75 | 0.75 |
| NACE-section S | 0.25 | 0.5 | 1 | 1 | 1 | 1 |
| Universities | 0.1 | | | | | |
| Working from home | maximum | | | | | |
| Household isolation | yes | | | | | |
| Work hour reduction | yes | | | | | |

Table 1 shows the timetable of NPI implementation in the baseline scenario mimicking the factual course of NPI implementation in the first wave of COVID19 in Germany in 2020. For kindergartens, schools and universities as well as the five NACE-sections the values reflect the degree of openness for each kind of location (cf. subsection Non-pharmaceutical interventions). For the NPI Working from Home the value *maximum* means the full utilization of industry-specific capacities to work from home. See S4 Table for a full list of home-office exploitation rates per NACE-division. For industry-specific values regarding the Work hour reduction see S3 Table.

To estimate the effects of specific NPIs, we simulate different contra-factual scenarios using timetables that differ in one or more aspects from the one used in the baseline model. It is important to note that each effect is the change in the number of cases if a given NPI had not been implemented while all other NPIs remain active (if not reported differently, e.g., the combined effects of education-related NPIs in the School_Kinder-scenario). Overall, the simulation experiment includes 9 counterfactual scenarios with differing timetables. The counterfactual scenarios and their deviation from the baseline scenario are listed in Table 2.

For each state and scenario, we simulate 100 days, starting with the day when a state had approximately 50 cases per 100,000 inhabitants. In each run of the simulation we collect the new infections per day, the cumulative cases on a day and the age of the infected agents. If the size of the agent population is exactly equal to 100000 cases, frequencies can be directly interpreted as cases per 100000 inhabitants. Otherwise, the infection frequencies are calculated by $I*100000/P$, where I is the number of infections and P is the actual population size in the

**Table 2. Manipulation of NPIs in counter-factual scenarios.**

| Scenario | Manipulation | Value | Baseline value |
|---|---|---|---|
| Open.Schools | Schools (degree of openness) | 1 | 0–0.3 |
| Open.Kinder | Kindergartens (degree of openness) | 1 | 0.1–0.6 |
| Open.Uni | Universities (degree of openness) | 1 | 0.1 |
| Normal.HomeOffice | Working from home (industry-specific use of hours worked f. home) | normal | maximum |
| No.Quarantine | Household-isolation | no | yes |
| Normal.Work.Hours | Work hour redu.; NACE-sec. G, I, P, R, S (degree of openness) | no; 1 | yes; 0.25–1 |
| Open.AllEduc | *Scenario combines Open.Schools & Open.Kinder & Open.Uni* | | |
| Open.Schools.Kinder | *Scenario combines Open.Schools & Open.Kinder* | | |

Table 2 shows the differences between the scenario-specific timetables and the baseline scenario timetables shown in column "Baseline value" and in detail in Table 1. The values in the baseline scenario represent the factual degree of NPI-implementation in the first wave of COVID19 in Germany 2020.

simulation. Therefore, all results can be interpreted as cases per 100000. Due to the many stochastic elements used in the simulation, each single run of the model returns different results, even when the same set of parameters is used. Thus, to obtain reliable results for a given set of parameters we run 60 repetitions of model runs and average the results.

## Calibration

Although the model includes relevant micro-level mechanisms of a pandemic and a representative agent population, we must perform some iterative fine-tuning as is often done in ABMs, i.e., calibrating the model to reproduce empirical data [38]. To calibrate the ABMs on infection frequencies in a given federal state, we determine the values of two parameters: the hourly probability an infected agent will infect another agent, and the time needed to quarantine a household after the first symptoms of a household member appear. Among the different simulated scenarios, the baseline scenario is the one whose goal is to represent the scenario that actually happened in reality and thus the one that should be able to reproduce the corresponding empirical data on infections. Therefore, we calibrated the model in the baseline scenario to the corresponding data on the cumulative infection frequencies per 100.000 inhabitants in a given state. We performed the calibration using the Latin Hypercube Sampling algorithm [38], which is implemented in the python package Spotpy [39]. For each model to be calibrated, we searched for the set of model parameters that generate simulated data with the smallest deviation to the corresponding daily empirical data on cumulative number of infections (data source: [40]). We stopped the calibration process when at least one set of parameters was found to generate simulated data with a fit of a root mean squared error less than 20. S6 Fig shows the parameter combinations sampled using Latin Hypercube Sampling and the resulting model fit using these parameter combinations in the baseline scenario. Table 3 shows the parameter combinations yielding the best model fit for each federal state.

Because the calibration of the model for a single federal state requires at least 180 simulation runs (each consisting itself of 60 internal replications and one replication takes approx. 20 minutes) due to varying parameter sets, each process of calibration was serialized on 192 cores provided by BW-HPC, a statewide computing and data infrastructure.

## Results

To derive the efficacy of NPIs in counterfactual scenarios, the first step is to ensure the predictive capability of the model by assessing its fit to the actual spread of COVID-19. The proposed model matches the empirical data of the daily number of cumulative cases for each of the four most affected federal states in Germany (Fig 1). Our model predicts the rapid rise of COVID-19 cases in the early stages of the pandemic as well as its subsequent decelerated growth in which the infection numbers stabilize. The model also simulates accurately the different levels of disease spread in the four states, which vary considerable by structure (cf. Supporting information).

After establishing that the model actually predicts the empirical spread of the Corona virus, we can turn to implementing counterfactual scenarios. We do that by removing each NPI that was at work in reality and examine the effect of its omission from the baseline scenario in the

**Table 3. Calibrated parameters.**

| Parameter | Baden-Wuerttemberg | Bavaria | Hamburg | Saarland |
|---|---|---|---|---|
| Infection probability per time step | 0.06213 | 0.0645 | 0.0502 | 0.04727 |
| Time steps from symptoms to household quarantine | 21 | 23 | 26 | 23 |

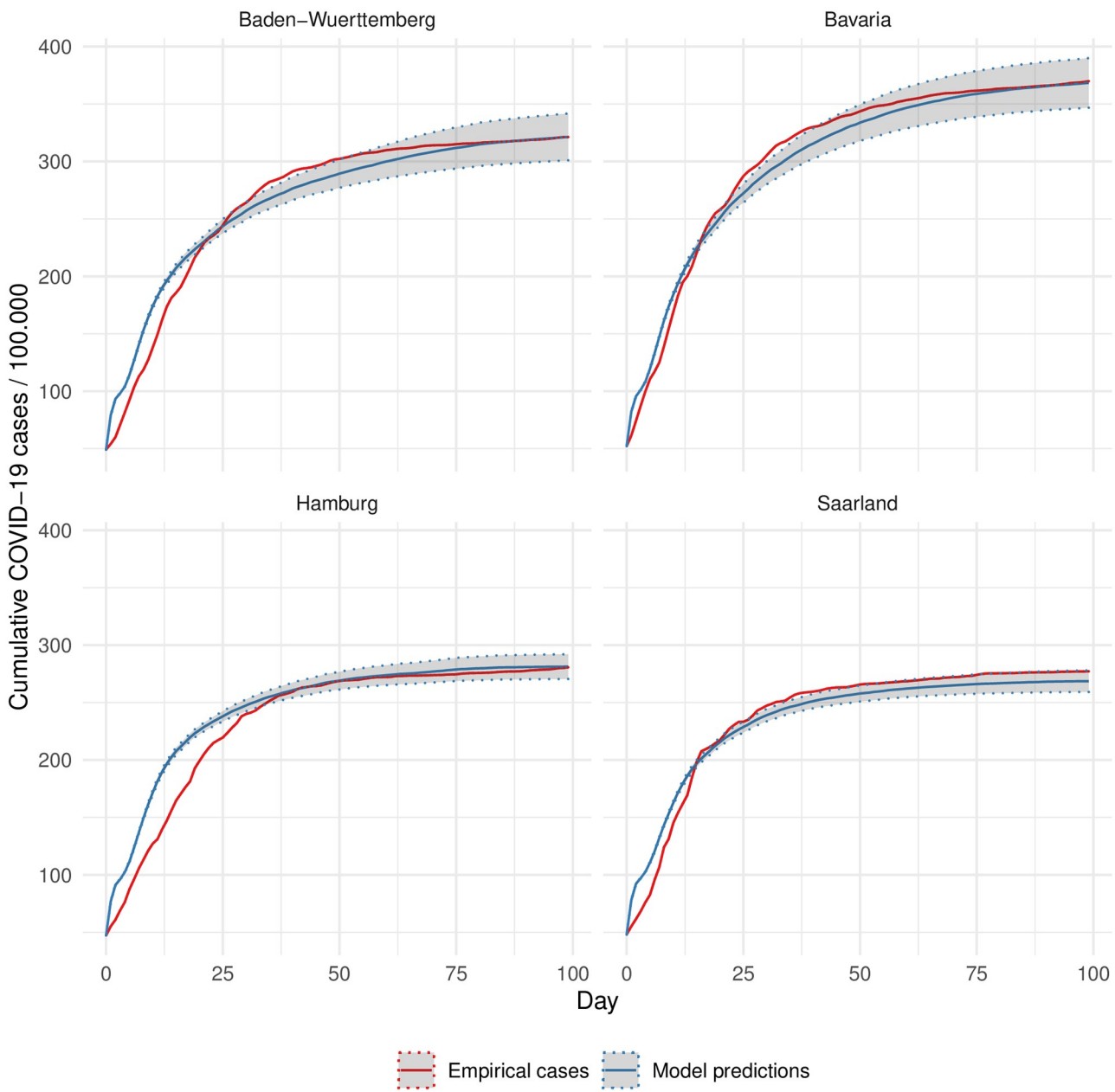

**Fig 1. Model fit by state.** The solid lines depict the cumulative number of reported COVID-19 cases and the average of cumulative infections across 60 simulation replications per state. Shaded areas represent the 95% confidence intervals of the averages. The root mean squared error of the model's predictions ranges between 8.414 for Bavaria and 16.56 for Hamburg.

simulation. Fig 2 reveals that decreasing the urge to quarantine infectious cases and reducing home office hours to regular percentages would cause the starkest increase across all implemented NPIs. While quarantine is, by a margin, the most important NPI in the two large territorial states (Baden-Wuerttemberg and Bavaria), it is a bit less important than increased home-office in the two smaller states. At the other end of the efficacy spectrum is the scenario in which there was no reduction of work hours. Across all states, it has almost no effect in our simulation if people would have worked normal hours. This small impact, e.g., compared to

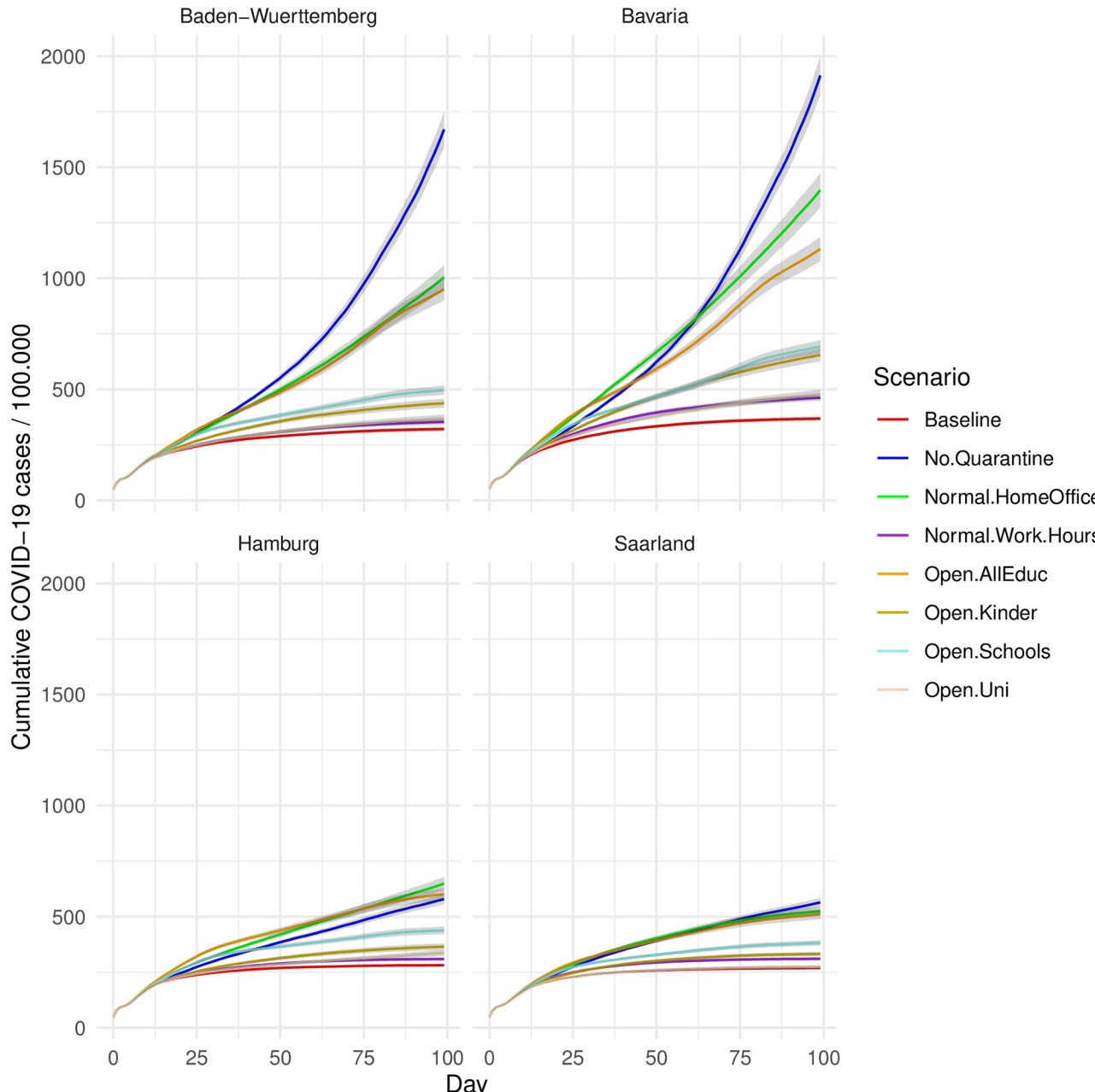

**Fig 2. Effect of counterfactual non-implementation of NPIs, by state.** Each line represents the average cumulative case numbers per scenario. While all NPIs considered are implemented in the baseline scenario to mimic the real scenario, the lines of the other scenarios show the estimated cumulative case numbers caused by the omission of one or more NPIs. For exact values of the average cumulative case numbers, standard deviations and confidence intervals on day 100, see also S5 and S6 Tables.

the impact of working from home, can be explained by the rather small empirical changes in working hours due to economic downturns. While working time reductions were close to zero in most industries, only hospitality industry (NACE-section I) and arts, entertainment and recreation (NACE-section R) experienced significant reductions of about 32.3% and 15.5%, respectively. For further details on the reduction of work hours per industry sector see S3

Table. In addition, the total closure of businesses and facilities is only selective and temporary in the baseline scenario, which can be seen in Tables 1 and 2.

Which education facilities may be kept running is a controversial topic in Germany, since it affects the lives of millions of children and parents. Turning to education-related NPIs, our model predicts a large increase of COVID-19 cases if all three types of institutions (kindergartens, schools, and universities) would re-open. In contrast to previous studies [1], our model also allows us to inspect the effect of each education facility separately. In the counterfactual scenarios presented in Fig 2, opening schools would have increased COVID-19 cases by several hundred (per day). Fig 3 disentangles the numbers further. It shows that opening all education-related institutions at the same time has a considerable effect (third largest overall, Fig 2), which exceeds the sum of the effects of each institution in all four regions. One possible explanation for this large effect of combined opening of educational institutions is that children who go to kindergarten often have siblings who go to school, and vice versa, as one can see in S9 Table. Thus, households with children could link two of the largest social network hubs (schools and kindergartens) in Germany, making it easy for the virus to spread to large segments of society that would otherwise be unreachable. Therefore, contagious effects seem at work which can be mostly attributed to the interplay of kindergarten kids and pupils (Open. Schools_Kinder in Fig 3). If comparing education-related NPI individually, we see that the effects of all three types are rather similar. Yet, opening universities has slightly lower effects than the other two, and opening schools causes the highest average effect in all four states.

In general, the opening of schools and kindergartens, both combined or each separately, seem to have notable effects on the case numbers due to two further points besides the linkage between pupils and kindergarten kids through households. Both pupils and kindergarten kids also typically live in relatively large households, which are breeding grounds for virus transmission (cf. S9 Table). In addition, those households not only seem to provide a link between pupils and kindergarten kids, but also provide a link between these two groups and the world of working adults, as S9 Table shows. On the other side, those household compositions can also explain the minor importance of universities regarding the number of infections. University students live in households of smaller average size compared to the households of other people. In addition, university students rarely live together with school kids or kindergarten kids, thus the contacts between those groups are limited (cf. S9 Table). Furthermore, in our sample the frequency of university students is much smaller than kindergarten kids or pupils (cf. S8 Table).

Besides considering NPIs separately, using an ABM with representative survey respondents as agents allows us to trace infections by age group. Age constitutes one of the main factors to identify risk groups [20]. In general and for each NPI, the mean age of infected agents decreases over time (Fig 4). This reflects recent findings using rare empirical data on the age of infected [41]. Furthermore, Fig 4 details how each NPI affects different age groups. It reveals that the counterfactual removal of NPIs drive COVID-19 numbers through the infection of agents younger than 60. Across all states, the scenarios in which schools re-open and pre-pandemic home office rates are attained lead to the lowest proportion among the elderly risk-group. In contrast, re-opening the universities yields a higher proportion of infections among people older than 60, although the vast majority of infections pertains younger people and the absolute numbers of COVID-19 caused by re-opening universities are rather low (Fig 2).

## Discussion

Overcoming the temporal paradox of deciding whether the implementation of a particular NPI is decreasing the spread of COVID-19 before knowing its actual effect is crucial for

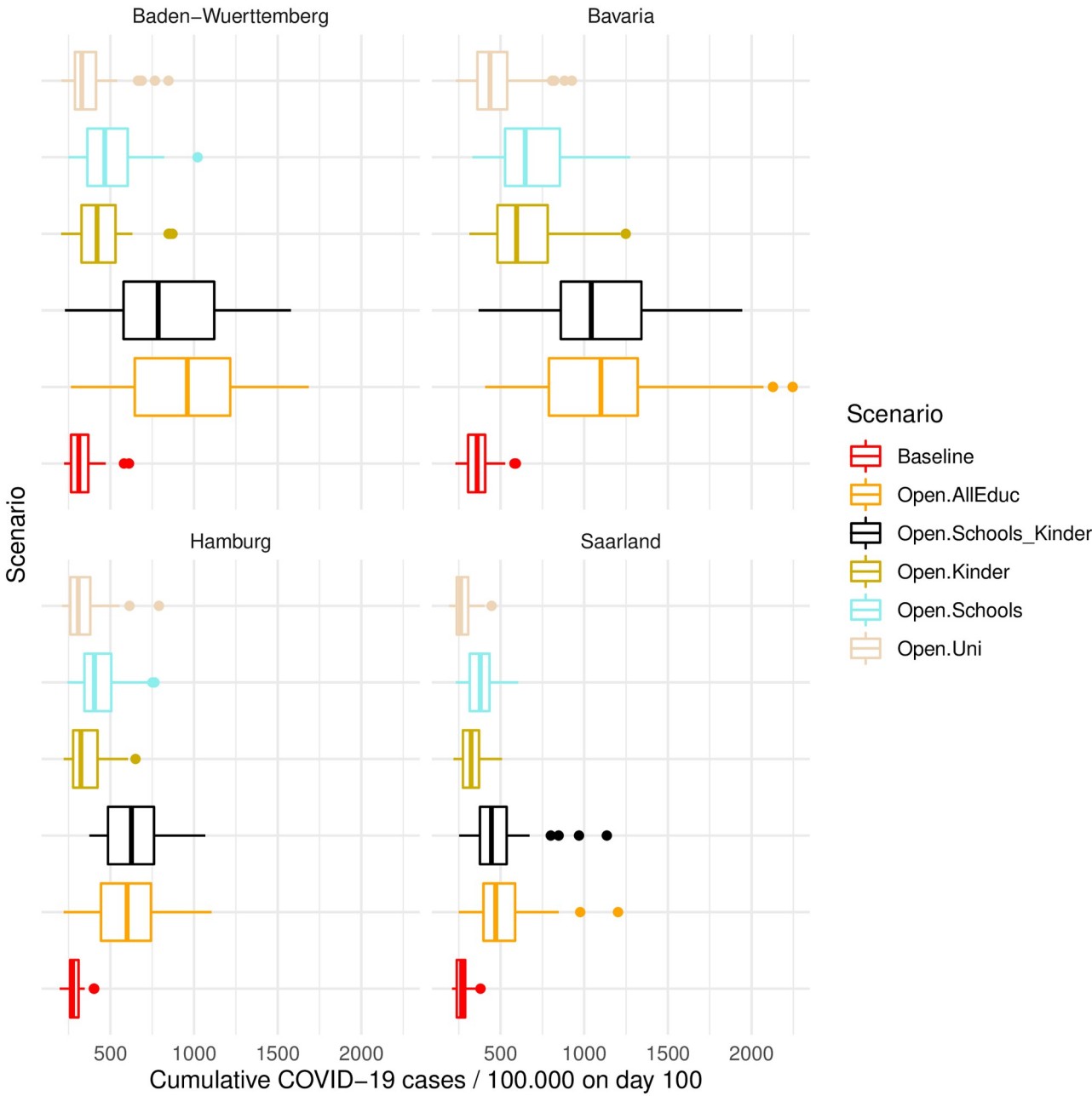

**Fig 3. Cumulative infection numbers on day 100 for each counterfactual non-implementation of education-related NPIs, by state.** Boxplots depict the distribution of all replications. For exact values of the average cumulative case numbers, standard deviations and confidence intervals on day 100, see also S5 and S6 Tables.

policy-makers and researchers alike. Testing such counterfactual what-if scenarios could improve the basis for decision-making considerably and help to decide which NPIs to implement. Our approach illustrates a way to estimate the effectiveness of otherwise unobservable scenarios by simulating agents with attributes taken from a large survey representing about 27 million people. In so doing, our study generates insights on the effectiveness of several widespread NPIs (quarantine, home-office, closing universities, schools, and kindergartens,

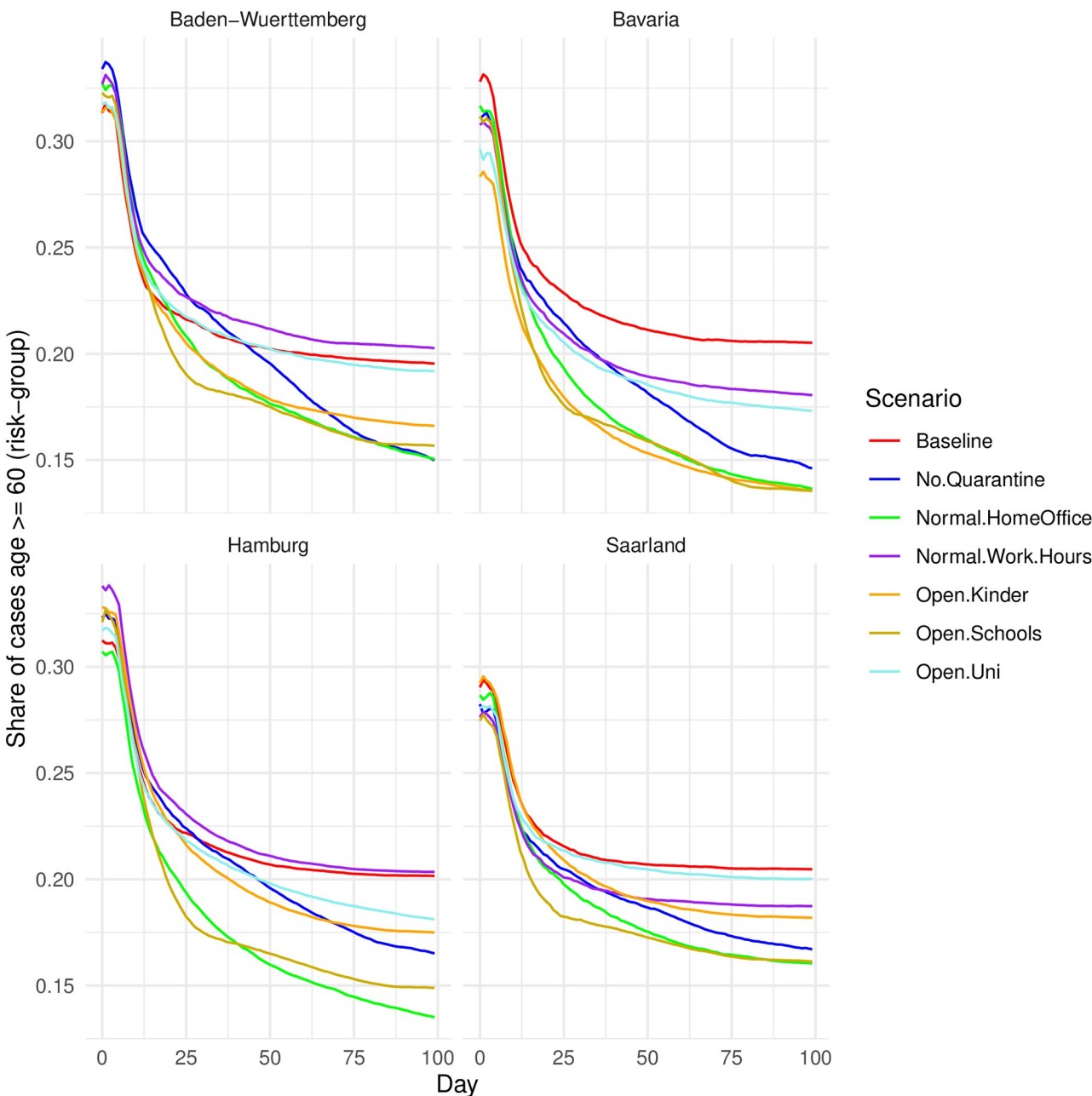

**Fig 4. Share of infected agents belonging to the risk-group (age $\geq$ = 60), by state.** Lines represent the share of cumulative cases in the risk group for each NPI over all simulation runs. To facilitate readability, a ribbon of confidence intervals is omitted. Figures including the CIs are available upon request.

reduced work hours) in four federal states in Germany. Going beyond previous studies, our approach enables us to disentangle the effect of individual NPIs and, hence, provides information for policy makers who need to balance the trade-off underlining each NPI between individual and economic freedom on the one side, and prevention of disease spread on the other.

It can be shown that focused quarantining of infected agents and their household members (i.e., close contacts) is the most effective NPI. Governments should therefore make every effort

to increase their ability of case detection and contact tracing [4, 5]. The second largest effect is the home office scenario. It can be shown that an increased usage of home office capacities is a very effective measure to reduce the number of new infections, which is in line with observational studies [42]. This result is particularly important, because although there was a considerable increase of home office usage in the beginning of the pandemic [43], both employers and employees still refuse to comprehensively rely on home office in the long term [44].

The most controversial NPI is the closure of educational facilities. In contrast to many observational studies [1, 16, 45], we are able to estimate a separate effect for each education-related NPI. Our simulations suggest that the separate re-opening of each institution yields rather low infection numbers. However, combined openings lead to multiple infections, in particular, re-opening kindergartens and schools at the same time increase COVID-19 cases considerably. Therefore, alternating periods of open educational institutions could represent a promising way to maintain low infection numbers while sustaining institutional functions.

A key characteristic of COVID-19 is its elevated mortality for older people and the subsequent importance of demographics [20]. Taking into account social structures by using agents with survey-characteristics and in accordance with a very recent study for the USA [41], our findings reveal that younger age (less risk-prone) groups sustain the spread of COVID-19 when the fore-mentioned NPIs are omitted. Furthermore, simulations suggest that the removal of NPIs affect age groups differently. Considering age adds an important puzzle piece how to select NPIs, i.e., it allows to distinguish the impact of NPIs in regard to its overall effects *and* group-specific effects for the people who are most vulnerable to COVID-19.

Like other simulation studies, our analysis has several limitations. First, our model does not contain a representation of contacts between households or agents that do not take place in one of the implemented locations, e.g., contacts with relatives or friends. Second, our model does not contain locations were people meet during spare time. However, the influence of both points is mitigated because private contacts and leisure activities were reduced to a minimum in the observed period in Germany. Having said that, care should be taken when applying the results to other countries or other times. For instance, while in our model elderly and retired people are somewhat protected against COVID-19 outbreaks among younger people because they often live in separate households and do not have explicit contact with relatives, this might not be the case in countries with different household compositions. Finally, the SOEP survey does not contain information on respondents' social network, hence, we could not consider this important transmission channel.

A further limitation is that we do not model changes of people's behavior. For example, we do not take into account the change in caution about infection risks in daily life or the obligation to keep distance and wear masks inside buildings. Also beyond the scope of this paper is the inclusion of external influences. Instead, we simulate a closed system. After the initial infections at the beginning of a run no infections occur due to agents coming home from outside, e.g., commuters or tourists.

As many other simulation studies, we had to assume some parameter values due to a lack of empirical data. However, we conducted a series of robustness checks for the most important of those parameters (cf. S1–S5 Figs). Sensitivity analyses suggest that the results are robust to changes in the parameters and, hence, to potential errors in assumed values.

Despite those limitations, we provide a reproducible model based on empirical agents which can be adopted and/or extended. For instance, the current model including above NPIs could be applied on any states or regions. Researchers would only need to calibrate the model with subsequent empirical data on COVID-19 cases and survey data. To represent ongoing vaccination efforts, the model could also be modified so that vaccinated agents have lower probabilities to transmit the virus. In addition, researchers could extend the existing model

with alternative NPIs which may be relevant for a specific region. The simulated world can be amplified by many other meeting places in which certain persons come into contact.

Despite all efforts and increasing availability of vaccinations, it appears that COVID-19 will accompany our societies for some time to come [46]. Thus, the demand for data-driven epidemiological models including realistic social structures and demography will stay high. Our model is an instance how to incorporate survey-based populations and test counterfactual scenarios. The development and improvement of such models and the timely availability of more accurate data was maybe never more important than in the current global pandemic.

## Supporting information

**S1 Table. Values and data sources of simulation parameters.** * At the time of conducting our simulation experiments, we used the most recent values listed in the latest version of Covasim and the associated article. Unfortunately, these values have changed (slightly) in the final release of Covasim. Nevertheless, we do not expect these changes to have a significant impact on our model results [34, 36, 47, 48].
(PDF)

**S2 Table. Descriptive statistics on federal states [34, 49, 50].**
(PDF)

**S3 Table. Reduction of work hours per NACE-section.** S3 Table gives the percentages to which the work hours were reduced in the second quarter (April, May, June) of 2020 as compared to 2019 per NACE-section. We assume that the differences between 2020 and 2019 are mainly caused by the COVID-19 pandemic. The values are taken from [51].
(PDF)

**S4 Table. Amount of working from home per NACE-divisions.** The column "Normal" gives the share of people frequently working at home office per NACE-division. The column "Maximum" gives the share to which the work could be done in home office i.e. the maximum capacity of working from home per NACE-division. We interpret all values as share of work hours per NACE-division. We assume that in the total lock- and shutdown in the first wave of COVID-19 in Germany 2020 the full capacity of working from home was exploited in each NACE-division. All values are taken from [35].
(PDF)

**S5 Table. Case numbers on day 100.** S5 Table shows the cumulative number of infected agents per 100,000 agents after 100 simulated days, averaged over 60 replications per state and scenario, along with the standard deviation and 95% confidence interval.
(PDF)

**S6 Table. Case numbers on day 100 as the difference to baseline scenario.** S6 Table shows the cumulative number of infected agents per 100,000 agents after 100 simulated days, averaged over 60 replications per state and scenario, as the difference from the corresponding baseline scenario, along with the standard deviation and 95 percent confidence interval.
(PDF)

**S7 Table. Infected agents by age-group.** S7 Table shows the proportion of each age group in the cumulative number of infected agents on simulation day 100.
(PDF)

**S8 Table. Share of vocational statuses per federal state.** S8 Table shows the proportion of different vocational statuses per federal state in the population of agents. "None/other" are those

agents who are neither working, studying, or of kindergarten or school age. "Working" means any agent who works more than zero hours per day and is not enrolled at a university.
(PDF)

**S9 Table. Household composition by vocational status and state.** S9 Table shows for each vocational status the average number of agents living in the same household by their vocational status. For instance, on average, an agent classified as a kindergarten child in Baden-Wuerttemberg lives in a household that contains 1.53 kindergarten children, 0.63 schoolchildren, 0.03 students, 1.4 employed persons, and 0.58 other persons. "None/other" are those agents who are neither working, studying, or of kindergarten or school age. "Working" means any agent who works more than zero hours per day and is not enrolled at an university.
(PDF)

**S1 Fig. Robustness check: Hours at kindergarten.** Compares the number of COVID-19 infections in the baseline scenario across different values for the daily hours at kindergarten. The value in the center is the one used in the main analysis.
(TIF)

**S2 Fig. Robustness check: Hours at school.** Compares the number of COVID-19 infections in the baseline scenario across different values for hours at school. The value in the center is the one used in the main analysis.
(TIF)

**S3 Fig. Robustness check: Hours at university.** Compares the number of COVID-19 infections in the baseline scenario across different values for hours at university. The value in the center is the one used in the main analysis.
(TIF)

**S4 Fig. Robustness check: Number of colleagues.** Compares the number of COVID-19 infections in the baseline scenario across different values for number of colleagues. The value in the center is the one used in the main analysis.
(TIF)

**S5 Fig. Robustness check: Maximum number of students per university.** Compares the number of COVID-19 infections in the baseline scenario across different values for the maximum number of students per university. We used 100,000 in the main analysis, which means that there is only one university per simulated world.
(TIF)

**S6 Fig. Sampled parameter space.** S6 Fig shows the tested combinations of the two calibrated parameters and the respective model fit when using the combination in the baseline model. The parameter combination that gives the best model fit is indicated by the cross.
(TIF)

## Acknowledgments

We are deeply grateful for helpful comments made by Lisa Schöllhammer, Uwe Remer and André Bächtiger, which greatly improved this manuscript. In addition, we thank the state of Baden-Wuerttemberg for providing us with computing capacity through bwHPC.

## Author Contributions

**Conceptualization:** Marius Kaffai, Raphael H. Heiberger.

**Data curation:** Marius Kaffai.

**Investigation:** Marius Kaffai, Raphael H. Heiberger.

**Methodology:** Marius Kaffai, Raphael H. Heiberger.

**Software:** Marius Kaffai.

**Validation:** Marius Kaffai, Raphael H. Heiberger.

**Visualization:** Raphael H. Heiberger.

**Writing – original draft:** Marius Kaffai, Raphael H. Heiberger.

**Writing – review & editing:** Marius Kaffai, Raphael H. Heiberger.

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
