## [Decision Letter · Decision Letter 0]

13 Aug 2021

PONE-D-21-14064

Modeling non-pharmaceutical interventions in the COVID-19 pandemic with survey-based simulations

PLOS ONE

Dear Dr. Kaffai,

Thank you for submitting your manuscript to PLOS ONE. Your paper has been carefully reviewed and was considered positively, overall. However, some aspects need further consideration and should be integrated with additional content. Therefore, we invite you to submit a revised version of the manuscript that addresses the points raised during the review process.

In particular, two are the aspects that need particular attention:

Model description: Reviewer #3 has noted that the model should be described more precisely and with further details in order to be completely clear to the reader. We invite authors to expand the description.Figure explanation and discussion: All reviewers have noted that figures are not completely clear, either in the discussion or in the caption. We invite authors to revise all figures, their captions, and the discussion of their content according to the comments of all reviewers.References: Pay specific attention to the following Journal Requirements, pointing to a possible problem with references to retracted articles. References should be carefully checked before resubmission, each one should have a clear rationale for being cited. All changes in the references should be reported in the rebuttal letter to inform the reviewers. 

Minor comments have been also provided and require attention as well.

Overall, the paper has merits and is close to the quality needed for publication, which justifies the minor revision status. This however does not imply that the revision could be limited to few superficial fixes. All comments raised by the reviewers and the journal's editors should be specifically addressed, each one needs a comment explaining how the authors decided to proceed, and edits in the manuscript should be appropriate to the comment raised.  

We look forward to receiving your revised manuscript.

Kind regards,

Marco Cremonini, Ph.D.

University of Milan

Academic Editor

PLOS ONE

1. Please ensure that your manuscript meets PLOS ONE's style requirements, including those for file naming. The PLOS ONE style templates can be found at https://journals.plos.org/plosone/s/file?id=wjVg/PLOSOne_formatting_sample_main_body.pdf and https://journals.plos.org/plosone/s/file?id=ba62/PLOSOne_formatting_sample_title_authors_affiliations.pdf.

4. PLOS requires an ORCID iD for the corresponding author in Editorial Manager on papers submitted after December 6th, 2016. Please ensure that you have an ORCID iD and that it is validated in Editorial Manager. To do this, go to ‘Update my Information’ (in the upper left-hand corner of the main menu), and click on the Fetch/Validate link next to the ORCID field. This will take you to the ORCID site and allow you to create a new iD or authenticate a pre-existing iD in Editorial Manager. Please see the following video for instructions on linking an ORCID iD to your Editorial Manager account: https://www.youtube.com/watch?v=_xcclfuvtxQ.

Additional Editor Comments (if provided):

Reviewers' comments:

Reviewer's Responses to Questions

**Comments to the Author**

1. Is the manuscript technically sound, and do the data support the conclusions?

Reviewer #1: Yes

Reviewer #2: Yes

Reviewer #3: Partly

2. Has the statistical analysis been performed appropriately and rigorously? 

Reviewer #1: Yes

Reviewer #2: Yes

Reviewer #3: N/A

3. Have the authors made all data underlying the findings in their manuscript fully available?

Reviewer #1: Yes

Reviewer #2: Yes

Reviewer #3: Yes

4. Is the manuscript presented in an intelligible fashion and written in standard English?

Reviewer #1: Yes

Reviewer #2: Yes

Reviewer #3: Yes

5. Review Comments to the Author

Reviewer #1: In this manuscript, the authors model the effects of non-pharmaceutical interventions to investigate the spread of COVID-19 for some states in Germany, using an agent-based model with a heterogeneous population based on survey data from the German Socio-Economic Panel. From the results of the simulations, they found that some scenarios, as quarantine and home-office, are the most effective measures to reduce COVID-19 cases. They also investigate and evaluate the effect of some NPIs on the number of COVID-19 cases in the risk group (age > 60 years). The research is interesting and presents a model that can be useful to assist in decision-making regarding COVID-19, a current health problem of great concern. The manuscript is well written, the methodology is sufficiently detailed and, the conclusions are supported by the results found. I consider that the article is scientifically sound and suitable for publication in PLOS ONE. I have a few comments nonetheless:

1. It is clear from the text that Figure 1 is the cumulative number of COVID-19 cases, but maybe the authors can make this clear in the caption or y-axis of the figure to avoid doubts. Does Figure 2 represent accumulated cases as well?

2. Perhaps the authors can explore the results of Figure 4 a little further. Are the differences in the behavior of the curves in the four states linked to demographic variation? Could the finding "the scenarios in which schools re-open and pre-pandemic home office rates are attained lead to the lowest proportion among the elderly risk-group" be influenced by the fact that the model does not consider contacts with relatives or friends (as children transmitting to their grandparents, for example)? (I understand that this would be reduced in the case of severe isolation measures, but it is an important point if the model is used in places where children live or stay with older relatives, as in some underdeveloped countries).

3. In the caption of Figure 4, the title says "Number of infected agents" while the legend says "share of cases". I suggest that authors standardize the name so that "number" is not confused with the absolute number. Also, perhaps the authors could consider presenting the percentages infected for each age group, not only for >=60 (maybe in a supplementary table with the final infected number or percentage), but this is just a suggestion.

4. Perhaps the authors might consider changing the yellow color in Figs 2 and 4, which can be difficult to see depending on the readers’ screen.

5. On page 6, line 302, if I am not mistaken, it seems like “Figures 5-9 in SI” means Figures S1-S5.

Reviewer #2: In this paper, “Modeling non-pharmaceutical interventions in the COVID-19 pandemic with survey-based simulations”, the authors aim to develop an agent-based model concerning the “what-if” fundamental question in implementing the NPIs’ during the COVID-19 pandemic.

The topic is interesting and the document is well presented. While the results trends are illustrative enough in the figures, I suggest to add a table which contains the standard deviation, mean and CI for each intervention removal's configuration. These tables, particularly for figure 3 and 4, make the comparison among baseline and other configurations more clear.

From my point of view the submitted paper is suitable for publication in PLOS ONE in the presented form.

Reviewer #3: The paper presents an application of the Agent-based modeling approach to study the efficacy of Non-pharmaceutical interventions (NPIs) following the outbreak of Covid-19. The authors consider a basic SEIR model and couple it with survey data to build a model where they can effectively track and modify the daily routines of the agents.

The approach adopted in the paper is novel and is a promising avenue of further research in agent-based models. While agent-based models have had success as diagnostic tools and have been able to provide great insight into many systems of interest, convincing applications that integrate real-world micro-level data have been few and far between. By basing their model on data from surveys on the behavior and routines of the people from 4 provinces of Germany, the paper is extremely important towards ABMs becoming more widespread as a modeling paradigm.

Having said that, the paper in question does suffer from a certain number of deficiencies, which undermine the message and the importance of the work. Let me begin by some general comments before providing specific comments:

1. A major shortcoming of the paper is that it isn't clear what the model actually is. Care should be taken to provide a detailed schematic of the various parts of the dynamical model and the different sections of the population considered along with a more detailed description of the NPIs.

2. Furthermore, reading the figures is a bit difficult since the baseline scenario is not the baseline "no-intervention" scenario (as one would intuitively expect) but with all NPIs in place. it could be better to relabel the individual curves so that "no-quarantine" becomes the baseline "no-intervention" scenario.

3. The authors provide a link to the repository where the code for the model is available. While laudable, it is difficult to know where and how to begin using the code since a complete Readme file has not been provided. Given the ever-growing importance of computer simulations, especially in ABMs, it is important that end-users be provided with all the necessary information to be able to run and explore the results of the model. Since the computer code is an integral part of the research underlying this paper, it is important that the authors take the time to address this point.

Moving on to specifics:

1. Lines 193-194: Are there any parameter configurations, other than those presented in the paper that lead to similar results? If so, how dissimilar are these parameter configurations as compared to those presented in the paper?

2. Line 218: What economic sectors are the authors referring to here?

3. Line 219: There seems to be an apparent contradiction here: While establishing home office worker proportions to pre-pandemic times leads to a marked increase in the number of cases, it is strange to see that people working "normal hours" (and thus presuambly going to work) doesn't increase case numbers. Why is this the case?

4. Fig 4: Why is it that the "no quarantine" scenario also leads to the share of cases in the "at-risk" population to reach levels similar to those of "open schools" and "normal home office"?

5. Fig 4: Caption in the body of the paper reads "Number of infected agents" while the figure itself plots the "share of the at-risk population".

Finally, at many points in the paper, the authors state the results of the simulations without providing any explanation as to what the probable causes could be. These are detailed below:

1. Line 216: Why is it the case that quarantine is a less important NPI in the two smaller states?

2. Lines 228-229: The authors write that the consequences of opening up all educational institutions are very different than if only one type of educational institution is opened? Why is this the case? Perhaps the survey data should be able to throw some light on the question.

3. Line 247: Why is it the case that the opening up of universities seems to play only a small role in the total number of cases?

A few typographical errors

1. In the abstract, line number 5 "researcher" should read "researchers".

2. Line 12: "decision-maker" should read "decision-makers".

3. Every instance when a quotation mark has been used must be corrected, for instance in lines 35 and 85.

6. PLOS authors have the option to publish the peer review history of their article (what does this mean?). If published, this will include your full peer review and any attached files.

Reviewer #1: No

Reviewer #2: No

Reviewer #3: No

---

## [Author Response · Author response to Decision Letter 0]

25 Sep 2021

The authors would like to thank the editor and reviewers for their time, attention and those valuable comments.

The comments significantly helped improving the manuscript, especially regarding the model description and output analysis.

We have addressed all the issues raised in this rebuttal letter. 

+++ Editor +++

 COMMENT 1.1 

In particular, two are the aspects that need particular attention:

Model description: Reviewer #3 has noted that the model should be described more precisely and with further details in order to be completely clear to the reader. We invite authors to expand the description.

Answer:

We have revised the section on Material and methods and added information on the design of the model and simulation experiments (cf. comment 4.2).

 COMMENT 1.2 

Figure explanation and discussion: All reviewers have noted that figures are not completely clear, either in the discussion or in the caption. We invite authors to revise all figures, their captions, and the discussion of their content according to the comments of all reviewers.

Answer:

We facilitated the readability of the figure captions and also followed reviewers advise on how to improve their appearance.

 COMMENT 1.3 

References: Pay specific attention to the following Journal Requirements, pointing to a possible problem with references to retracted articles. References should be carefully checked before resubmission, each one should have a clear rationale for being cited. All changes in the references should be reported in the rebuttal letter to inform the reviewers. 

We have checked all our references and have found no article retracted from a journal in our reference list.

Nevertheless, we have revised our references in regard to preprints.

Accordingly, we updated preprints that have been published in journals.

Preprints that have not yet been published in a journal and are not essential to this study have been removed as references.

Furthermore, we have added three more references.

Removed references:

Chao DL, Oron AP, Srikrishna D, Famulare M. Modeling layered

non-pharmaceutical interventions against SARS-CoV-2 in the United States with

Corvid. medRxiv. 2020; p. 2020.04.08.20058487. doi:10.1101/2020.04.08.20058487.

Preprints replaced by the journal publication:

Kerr CC, Stuart RM, Mistry D, Abeysuriya RG, Hart G, Rosenfeld K, et al.

Covasim: an agent-based model of COVID-19 dynamics and interventions.

medRxiv. 2020; p. 2020.05.10.20097469. doi:10.1101/2020.05.10.20097469.

Alipour JV, Fadinger H, Schymik J. My Home is My Castle - The Benefits of

Working from Home During a Pandemic Crisis: Evidence from Germany.

University of Bonn and University of Mannheim, Germany; 2020.

crctr224 2020 178.

Hunter PR, Colon-Gonzalez F, Brainard JS, Rushton S. Impact of

non-pharmaceutical interventions against COVID-19 in Europe: a

quasi-experimental study. medRxiv. 2020; p. 2020.05.01.20088260.

doi:10.1101/2020.05.01.20088260.

Added references:

Grimm V, Berger U, Bastiansen F, Eliassen S, Ginot V, Giske J, et al. Astandard protocol for describing individual-based and agent-based models.Ecological Modelling. 2006;198(1-2):115–126. doi:10.1016/j.ecolmodel.2006.04.023.

Grimm V, Railsback SF, Vincenot CE, Berger U, Gallagher C, DeAngelis DL,et al. The ODD Protocol for Describing Agent-Based and Other SimulationModels: A Second Update to Improve Clarity, Replication, and StructuralRealism. Journal of Artificial Societies and Social Simulation. 2020;23(2):7.

Destatis. Arbeitnehmerverdienste. 2. Vierteljahr 2020. Fachserie 16 Reihe 2.1.; 2020.

+++Reviewer 1 +++

 COMMENT 2.1 

In this manuscript, the authors model the effects of non-pharmaceutical interventions to investigate the spread of COVID-19 for some states in Germany, using an agent-based model with a heterogeneous population based on survey data from the German Socio-Economic Panel. From the results of the simulations, they found that some scenarios, as quarantine and home-office, are the most effective measures to reduce COVID-19 cases. They also investigate and evaluate the effect of some NPIs on the number of COVID-19 cases in the risk group (age > 60 years). The research is interesting and presents a model that can be useful to assist in decision-making regarding COVID-19, a current health problem of great concern. The manuscript is well written, the methodology is sufficiently detailed and, the conclusions are supported by the results found. I consider that the article is scientifically sound and suitable for publication in PLOS ONE. I have a few comments nonetheless:

Answer:

We thank reviewer 1 for the overall positive assessment of our work and the helpful comments to further improve the manuscript.

 COMMENT 2.2 

It is clear from the text that Figure 1 is the cumulative number of COVID-19 cases, but maybe the authors can make this clear in the caption or y-axis of the figure to avoid doubts. Does Figure 2 represent accumulated cases as well?

Answer:

We have changed the y-axis of Figure 1 and 2 to "Cumulative COVID-19 cases / 100.000" to highlight that cumulative case counts are reported in both figures.

In addition, we have revised the caption of the figures.

 COMMENT 2.3 

Perhaps the authors can explore the results of Figure 4 a little further. Are the differences in the behavior of the curves in the four states linked to demographic variation? Could the finding "the scenarios in which schools re-open and pre-pandemic home office rates are attained lead to the lowest proportion among the elderly risk-group" be influenced by the fact that the model does not consider contacts with relatives or friends (as children transmitting to their grandparents, for example)? (I understand that this would be reduced in the case of severe isolation measures, but it is an important point if the model is used in places where children live or stay with older relatives, as in some underdeveloped countries).

Answer:

We thank the reviewer for these very important issues.

Also one of the motivations of this project has been to link case numbers to demographic variation between different states, it turned out to be not feasible in the context of this article with its many scenarios and few considered states.

We approach this issue in future research in an update of the model together with a more specific research design.

Nevertheless, we have added paragraphs on page 10 linking the results of Figure 2 and 3 to demographic properties of all states, particularly household composition.

Furthermore, we think the reviewer is correct that the low proportion of infections among the elderly risk-group is partly caused by not explicitly considering contacts between households of relatives and friends.

While we assume this model design is appropriate for the early pandemic phase in germany 2020, the reviewer is right that this might be inappropriate if applied to other countries (or other times).

Thus, we have added multiple phrases addressing this issue on page 11.

 COMMENT 2.4 

In the caption of Figure 4, the title says "Number of infected agents" while the legend says "share of cases". I suggest that authors standardize the name so that "number" is not confused with the absolute number. Also, perhaps the authors could consider presenting the percentages infected for each age group, not only for >=60 (maybe in a supplementary table with the final infected number or percentage), but this is just a suggestion.

Answer:

We have changed the title of Figure 4 to "Share of infected agents belonging to the risk-group (age $>=$ 60), by state.".

Based on the suggestion, we have added Table S7, listing each age-group's share in the infected population.

 COMMENT 2.5 

Perhaps the authors might consider changing the yellow color in Figs 2 and 4, which can be difficult to see depending on the readers’ screen.

Answer:

Thank you very much for this remark.

We changed the color scheme and avoid yellow in all figures.

 COMMENT 2.6 

On page 6, line 302, if I am not mistaken, it seems like “Figures 5-9 in SI” means Figures S1-S5.

\\end{mdframed}

Answer:

That is absolutely correct.

We have corrected the respective passage accordingly.

+++ Reviewer 2 +++

 COMMENT 3.1 

In this paper, “Modeling non-pharmaceutical interventions in the COVID-19 pandemic with survey-based simulations”, the authors aim to develop an agent-based model concerning the “what-if” fundamental question in implementing the NPIs’ during the COVID-19 pandemic.

The topic is interesting and the document is well presented. While the results trends are illustrative enough in the figures, I suggest to add a table which contains the standard deviation, mean and CI for each intervention removal's configuration. These tables, particularly for figure 3 and 4, make the comparison among baseline and other configurations more clear.

From my point of view the submitted paper is suitable for publication in PLOS ONE in the presented form.

Answer:

We thank reviewer 2 for this positive evaluation of our manuscript and the valuable suggestion to include additional tables on the results of the simulation experiments.

Based on this comment, we have added Table S5 and S6 to the appendix, which contain the requested numbers.

+++ Reviewer 3 +++

 COMMENT 4.1 

The approach adopted in the paper is novel and is a promising avenue of further research in agent-based models. While agent-based models have had success as diagnostic tools and have been able to provide great insight into many systems of interest, convincing applications that integrate real-world micro-level data have been few and far between. By basing their model on data from surveys on the behavior and routines of the people from 4 provinces of Germany, the paper is extremely important towards ABMs becoming more widespread as a modeling paradigm.

Having said that, the paper in question does suffer from a certain number of deficiencies, which undermine the message and the importance of the work. Let me begin by some general comments before providing specific comments:

Answer:

We thank reviewer 3 very much for their careful evaluation of our manuscript, the detailed, critical and helpful comments, as well as for the appreciation of our work.

 COMMENT 4.2 

1. A major shortcoming of the paper is that it isn't clear what the model actually is. Care should be taken to provide a detailed schematic of the various parts of the dynamical model and the different sections of the population considered along with a more detailed description of the NPIs.

Answer:

Following Reviewer 3's advise, we have completely revised the section on "Material and methods".

The model description now starts with a description of all entities (subsection "Entities and scales"), followed by a description of the initialization process (subsection "Initialization"), and the schedule of the main processes (subsection "Main simulation loop") including the content of these processes (subsections "Activities" and "Disease process and virus transmission").

In two subsections ("Non-pharmaceutical interventions" and "Scenarios and model execution"), we have also added a detailed description of the NPIs and how the scenarios were designed.

The newly added Table 1 and Table 2 give an overview of the implementation of NPIs in the baseline scenario and how the counterfactual scenarios differ from the baseline scenario.

 COMMENT 4.3 

2. Furthermore, reading the figures is a bit difficult since the baseline scenario is not the baseline "no-intervention" scenario (as one would intuitively expect) but with all NPIs in place. it could be better to relabel the individual curves so that "no-quarantine" becomes the baseline "no-intervention" scenario.

Answer:

We appreciate this comment, yet the "no-quarantine"-scenario is not a scenario without any interventions, but only without quarantining households.

The extended description of NPIs and the baseline scenario in response to comment 4.2 should make this more clear in the revised manuscript.

 COMMENT 4.4 

3. The authors provide a link to the repository where the code for the model is available. While laudable, it is difficult to know where and how to begin using the code since a complete Readme file has not been provided. Given the ever-growing importance of computer simulations, especially in ABMs, it is important that end-users be provided with all the necessary information to be able to run and explore the results of the model. Since the computer code is an integral part of the research underlying this paper, it is important that the authors take the time to address this point.

Answer:

We have added a README file containing a step-by-step guide for setup and execution of the model.

The README file can be found via this link: https://github.com/mariuzka/covid19_sim#readme.

 COMMENT 4.5 

Moving on to specifics:

1. Lines 193-194: Are there any parameter configurations, other than those presented in the paper that lead to similar results? If so, how dissimilar are these parameter configurations as compared to those presented in the paper?

Answer:

We infer two parameters ("time from symptoms to quarantine", "infection probability") by calibrating the model to empirical infection data (cf. subsection "Calibration", p. 8). 

We tested a wide range of configurations of those two and take the combination with the best model fit.

To illustrate this process in greater detail, we added Fig S6 in the appendix. 

It clearly shows that the chosen combinations are no outliers. 

 COMMENT 4.6 

2. Line 218: What economic sectors are the authors referring to here?

Answer:

With 'certain economic sectors' we meant both the NACE-sectors that had to partially close their business as well as the overall work time reduction due to the economic down turn in the most affected branches.

The NACE-sectors affected by partial shutdown are I, R, G, P and S (as now listed in Table 1).

Which occupational fields are most affected by the overall reduction of working hours is now listed in Table S3.

The newly added description of the NPIs and the scenarios should now make this point clear as well.

Because we now find that the wording 'certain economic sectors' is to unspecific, we have changed the corresponding phrase to "At the other end of the efficacy spectrum is the scenario in which there was no reduction of work hours.", which can be found at page 9 of the updated version of the manuscript.

 COMMENT 4.7 

3. Line 219: There seems to be an apparent contradiction here: While establishing home office worker proportions to pre-pandemic times leads to a marked increase in the number of cases, it is strange to see that people working "normal hours" (and thus presuambly going to work) doesn't increase case numbers. Why is this the case?

Answer:

This contradiction can be explained by the fact that changes in working hours due to business closures and economic downturns tend to be small compared to changes in hours worked in the office due to the (industry-specific) exhaustion of the possibilities of working from home.

In the updated version of the manuscript one can see the changes in the hours worked at the work place due to business closure and economic downturn in Table 1 and Table S3, while changes due to an extended home-based work arrangement are shown in Table S4.

Furthermore, one has to consider that in the counterfactual scenario where no reduction of work takes place the instruction to work from home whenever possible is still in place.

Thus, the effect of the number of working hours is suppressed by the effect of the amount of hours worked from home.

Based on this comment, we have added a paragraph explaining this issue at page 9.

 COMMENT 4.8 

4. Fig 4: Why is it that the "no quarantine" scenario also leads to the share of cases in the "at-risk" population to reach levels similar to those of "open schools" and "normal home office"?

Answer:

We think this comment might be related to the ambiguous explanation of the baseline scenario in the prior version of the manuscript (see also comment 4.3). 

It should now be clear that the "no quarantine" NPI mentioned by Reviewer 3 is not the baseline scenario.

 COMMENT 4.9 

5. Fig 4: Caption in the body of the paper reads "Number of infected agents" while the figure itself plots the "share of the at-risk population".

Answer:

We have corrected this error.

In the revised version, the caption of Figure 4 now reads "Share of infected agents belonging to the risk-group (age>=60), by state.".

 COMMENT 4.10 

Finally, at many points in the paper, the authors state the results of the simulations without providing any explanation as to what the probable causes could be. These are detailed below:

1. Line 216: Why is it the case that quarantine is a less important NPI in the two smaller states?

Answer:

Unfortunately, providing a detailed explanation for all mechanisms at work is beyond the scope of our paper.

We focus on the development of a model that allows the inspection of counterfactual NPIs, resting on simulations with survey-based agents.

While investigating each underlying effect and its driving factor is, in general, possible with this model, the current paper is, in our view, sufficiently complex by providing the model description, model fit and a "ranking" of NPIs' efficacy. 

However, we added further explanations as suggested by Reviewer 3.

 COMMENT 4.11 

2. Lines 228-229: The authors write that the consequences of opening up all educational institutions are very different than if only one type of educational institution is opened? Why is this the case? Perhaps the survey data should be able to throw some light on the question.

Answer:

An analyses of the household compositions reveals that kindergarten kids and school kids often live together with siblings that also go to kindergarten or school.

Thus, if both institutions are open, those households provide an open path the virus can take reaching other societal areas.

In addition, those households are often relatively big and also include working parents.

In order to clarify this point in the manuscript, we have added paragraphs on page 9 and 10 as well as the tables S8 and S9.

 COMMENT 4.12 

3. Line 247: Why is it the case that the opening up of universities seems to play only a small role in the total number of cases?

Answer:

Universities seem to play a minor role in our model, because the share of students is small compared to other groups, students live in smaller households which rarely include kindergarten- or school-age kids.

The manuscript now contains a paragraph on page 10 explaining this issue.

 COMMENT 4.13 

A few typographical errors

1. In the abstract, line number 5 "researcher" should read "researchers".

2. Line 12: "decision-maker" should read "decision-makers".

3. Every instance when a quotation mark has been used must be corrected, for instance in lines 35 and 85.

Answer:

We have corrected the mentioned typographical errors.

---

## [Editor Report · Decision Letter 1]

13 Oct 2021

Modeling non-pharmaceutical interventions in the COVID-19 pandemic with survey-based simulations

PONE-D-21-14064R1

Dear Dr. Kaffai,

We’re pleased to inform you that your manuscript has been judged scientifically suitable for publication and will be formally accepted for publication once it meets all outstanding technical requirements.

Kind regards,

Marco Cremonini, Ph.D.

University of Milan, Italy

Academic Editor

PLOS ONE
---

## [Editor Report · Acceptance letter]

20 Oct 2021

PONE-D-21-14064R1 

Modeling non-pharmaceutical interventions in the COVID-19 pandemic with survey-based simulations 

Dear Dr. Kaffai:

I'm pleased to inform you that your manuscript has been deemed suitable for publication in PLOS ONE. Congratulations! Your manuscript is now with our production department. 

Kind regards, 

on behalf of

Dr. Marco Cremonini 

Academic Editor

PLOS ONE